# Progress in Avalanche Photodiodes for Laser Ranging

**DOI:** 10.3390/s25092802

**Published:** 2025-04-29

**Authors:** Zhenxing Liu, Ning An, Xingwei Han, Natalia Edith Nuñez, Liang Jin, Chengzhi Liu

**Affiliations:** 1Changchun Observatory, National Astronomical Observatories, Chinese Academy of Sciences, Changchun 130117, China; liizhenxing23@mails.ucas.ac.cn (Z.L.);; 2University of Chinese Academy of Sciences, Beijing 100049, China; 3Faculty of Exact, Physical and Natural Sciences, National University of San Juan, San Juan J5402, Argentina; 4School of Communication Engineering, Hangzhou Dianzi University, Hangzhou 310018, China

**Keywords:** laser ranging, photodetectors, high accuracy, long range detection

## Abstract

Laser ranging is a high-precision geodetic technique that plays an indispensable role in the field of geodynamics. Avalanche photodiodes (APDs) offer a series of advantages over other photodetector technologies, including photomultiplier tubes (PMTs) and superconducting single-photon detectors (SNSPDs). These advantages include high sensitivity, small size, high integration, and low power consumption, which have contributed to the widespread use of APDs in laser ranging applications. This paper analyses the key role of APDs in enhancing the accuracy and stability of laser ranging through the examination of application examples, including Si-APD and InGaAs/InP APD. Finally, based on the technological needs of laser ranging, the future development directions of APDs are envisioned, aiming to provide a reference for the research of photodetectors in high-precision and high-frequency laser ranging applications.

## 1. Introduction

Laser ranging is a high-precision, long-range space geodesy technology that is widely employed in the fields of satellite orbiting, determination of the Earth’s rotation parameters and laser time comparison [1,2]. As the deep space exploration project has progressed, the laser ranging system has undergone a series of upgrades, progressing from the first generation to the third. These upgrades have resulted in significant improvements in terms of system optimisation [3]. The earlier generations were characterised by a number of limitations, including their bulkiness, weight, high energy consumption, and reliance on manual operation. In contrast, the third generation has achieved miniaturisation, high accuracy, automation, and multi-functionality. Notable among these developments are the emergence of representative technologies such as satellite laser ranging (SLR), lunar laser ranging (LLR), and debris laser ranging (DLR), which are providing substantial technical support for the advancement of manned spaceflight, satellite navigation, and other significant national special projects. With the continuous development of laser ranging technology, the use of high repetition frequency, low energy lasers as signal sources, and high sensitivity single-photon detectors has become the mainstream choice in the industry. At present, single-photon detectors for laser ranging mainly include photomultiplier tubes (PMTs) [4], Geiger-mode APDs [5], and superconducting nanowire single-photon detectors (SNSPDs) [6]. Among them, PMT has the advantages of high gain, a large photosensitive area, and low dark count, but its larger size, lower quantum efficiency, high reverse bias requirement, and weak immunity to external magnetic field are not conducive to its array expansion. As an emerging single-photon detector, SNSPDs feature a wide spectral response, extremely low dark counts, and minimal time jitter. However, their cryogenic operation requirements significantly constrain their practical deployment [7]. In contrast, Geiger-mode APDs offer lower power consumption and better resistance to electromagnetic interference compared to PMTs, while avoiding the cryogenic cooling systems required by SNSPDs. These comprehensive advantages have enabled APDs to become the dominant choice for modern laser ranging systems. The continuous extension of the detection distance of laser ranging systems and the increasing number of observed target types have resulted in a reduction in the intensity of laser ranging echo signals to a level below that of a single photon. The performance of the photoelectric receiver has become a critical determinant of the detection accuracy and stability of the system. Therefore, based on the working principle of laser ranging, this paper analyses the key role and development status of avalanche photodiode in the field of laser ranging by analysing the application requirements of the laser ranging system on photodetectors. Furthermore, in view of the evolution of laser ranging technology, the prospective trajectory of laser ranging single-photon detection technology is delineated.

## 2. Laser Ranging

Laser ranging technology is primarily based on the time-of-flight principle. Depending on the manner of its implementation, time-of-flight measurement can be classified into two distinct categories: direct and indirect. Direct time-of-flight ranging technology employs a precise calculation of the distance between the target and the measurement point, determined by the time interval between the emission of a laser pulse and its reflection back from the target. Indirect time-of-flight ranging inverts the measured distance based on other characteristics of the signal, including linear FM laser ranging and phased laser ranging [8,9,10]. In consideration of the technical challenges and financial investment required, direct time-of-flight ranging is a prevalent and effective method of detection in contemporary laser ranging applications.

### 2.1. Impulse Time-of-Flight Ranging Method

The time-of-flight (TOF) method is a distance measurement technique that employs the time difference Δ*T* between transmitted and received pulses to determine the distance between a target and a system. In accordance with the Equation (Equation 1), the distance between the observatory and the target is as follows:(1)$$R=\frac{c\cdot \Delta T}{2}=\frac{c\cdot (t_{2}-t_{1}-T)}{2}$$
where $t_{1}$ represents the moment of laser emission, $t_{2}$ denotes the echo reception moment, *T* signifies the delay induced by systematic, atmospheric and relativistic effects, satellite center-of-mass shift and other errors, and *c* is the propagation speed of light.

### 2.2. Time-Correlated Photon-Counting Method

The number of echo photons has a direct impact on the accuracy and reliability of laser ranging. With the continuous expansion of the detection distance of laser ranging technology and the diversification of observation target types, the laser echo photon intensity is gradually approaching the limit threshold of photon detection, i.e., the single-photon level, and the signal-to-noise ratio of the ranging system is also reduced, which puts forward higher requirements for the detection accuracy and reliability of laser ranging system. Table 1 presents the observed target echo photon counts at varying distances, as calculated using the radar correction equation and based on the parameters of the SLR system at the Changchun station.

The advent of time-correlated single photon-counting (TCSPC) technology has pushed the detection sensitivity of laser ranging systems to new heights. Unlike traditional linear laser ranging, the essence of TCSPC laser ranging technology is a ranging method based on the cumulative detection of weak photon signals, i.e., by repeatedly measuring the same position of the target, the detection of echo waveforms containing a large number of photon signals in the linear system is transformed into the counting of single echo photon events, and the target signal is extracted by using multiple accumulations of photon events to improve the detection probability. The technology is characterised by a high density of laser echo data and fast target search. By increasing the repetition rate of laser ranging, the accurate measurement of distant targets can be achieved even at low laser energy levels.

For the existing telescope system, TCSPC high-frequency laser ranging technology is a key means of improving detection capability. At present, international technological breakthroughs in ultra-high frequency SLR above 100 kHz have been achieved [11], laying the foundation for space debris laser ranging technology in the range of 100 KHz to 1 MHz. Figure 1 illustrates the distribution of photon counts based on time variation. It can be seen that the TCSPC technique only measures the presence or absence of a signal within each signal continent and does not record the intensity of the signal. During a signal period, if a photon is measured, 1 is added to the time storage unit corresponding to the arrival time of that photon, and if no photon is measured, it is not recorded. After repeating the measurement several times, the signal waveform can be recovered in the memory cell.

### 2.3. Laser Ranging System

The laser ranging system mainly consists of a computer control system, single-photon detector, telescope, laser, timing system, etc. The block diagram of the SLR ranging system is shown in Figure 2.

The process of laser ranging is as follows: First, the computer downloads the satellite’s orbit prediction, and the control servo system accurately tracks the satellite. The computer control system transmits pulses to the satellite according to the forecast and records the pulse transmission time $t_{1}$, the laser pulse is reflected by the satellite’s corner reflector to the receiver system, the detector receives the return photons and records the return time $t_{2}$, and finally, the computer pairs the main wave and the return time and completes the data processing to generate the laser ranging data product.

In satellite laser ranging (SLR) systems, the precise alignment and nanosecond synchronisation of the laser, detector, and telescope are the core elements that determine the efficiency and accuracy of the system. At the optical integration level, the micro-arcsecond alignment of the laser transmitter optical axis and the telescope receiver optical axis ensures the efficient coupling of the return signal, while optical path offset or thermal deformation will directly lead to signal attenuation or even loss; in terms of time synchronisation, the master clock triggers the laser pulse emission through the fibre-optic network synchronously and controls the time gating window of the single-photon detector, and accurately labels the flight timestamps, so that if the synchronisation deviation is more than the detector’s dead time, it will trigger the photon-counting noise multiplication and time measurement loss. The optimisation of photon detection is particularly critical: The use of an avalanche photodiode (APD) with an ultra-low dark count rate or a superconducting nanowire single-photon detector (SNSPD, >90% detection efficiency), combined with a narrow-band filter and spatial filtering, suppresses the background noise down to minimum level during the daytime; synchronised activation of the detector’s nanosecond time window following the emission of the laser pulse, in conjunction with multi-pulse accumulation, constant-ratio timing screening, and TCSPC technology, facilitates millimetre-level ranging accuracy even in the presence of weak signals at the single-photon level.

The application of single-photon detectors not only reduces the limitation of laser emission power and expands the range of laser selection, but also simplifies the design of the driving power supply and improves the system efficiency. In addition, the application of a high-sensitivity single-photon detector makes the small-diameter telescope able to receive signals effectively, significantly reduces the volume and weight of the system, provides convenience for the system to integrate more lasers and detectors, and greatly promotes the development and application of laser ranging technology.

## 3. Avalanche Photodiode

The avalanche photon diode (APD) is a type of photodiode that utilises the avalanche multiplication effect of semiconductor materials to achieve the highly sensitive detection of optical signals. Compared with conventional photodiodes, APDs add a high electric field region to their internal structure, which enables the photogenerated carriers generated by incident photons to trigger an avalanche effect when passing through this region, resulting in a large number of electron–hole pairs. This multiplication effect significantly increases the sensitivity of the APD, allowing it to effectively detect optical signals even in low-light environments. A typical avalanche diode structure is shown in Figure 3.

Research into APDs began in the 1960s. Initially, the APD bias voltage was very high and only semiconductor materials such as Si and Ge were used [12,13]. In the 1980s, APDs were developed from III-V semiconductor materials, represented by GaAs materials, and a variety of APD designs appeared, including table-top structures, ion-implanted structures, and heterojunction APDs. By the mid-1990s, Si-based APDs were gradually being commercialised, and PerkinElmer developed APDs with photon detection efficiencies (PDE) as high as 70% for 600 nm wavelengths, with a dark count rate of 1 kHz and relatively low bias voltages (200 V–600 V). The C30734E series of Si APDs manufactured by Excelitas Technologies, Canada, has a good spectral response in the wavelength range of 400 to 1000 nm, with a gain of up to 100%. The IDQ idquantique company developed the ID100 series, which features a 20/50 μm detection surface and ultra-low dark counts. However, the detection efficiency of this series is only 7% @ 800 nm. Subsequently, the ID120 series was introduced, which has a 500 μm detection surface and, due to its feed-through structure, can enhance the quantum efficiency to 80% @ 800 nm. The 21st century has witnessed rapid development in optoelectronics and semiconductor technology, leading to the creation of new APDs with new structures such as quantum dots and quantum wells, etc. [14,15,16,17]. At the same time, the rapid progress in integrated optoelectronics technology has made it possible to integrate APDs with other optoelectronic devices such as lasers, modulators, waveguides, etc., which has significantly improved the accuracy and reliability of the system detection system.

APD operating modes are mainly divided into linear mode and Geiger mode. In linear mode, the gain of the APD is controlled within a certain range to maintain the linearity of the output signal. The gain of the APD increases gradually as the voltage increases. When the operating voltage exceeds the breakdown voltage, carrier generation continues, avalanche breakdown occurs, and the APD enters Geiger mode, also known as the SPAD (single-photon avalanche diode). An avalanche photodiode (APD) operating in linear mode has an output current that is linearly proportional to the incident light intensity. Although this mode has a high response sensitivity, its single-photon signals are easily swamped by noise and are usually suitable for detecting optical signals with photon counts on the order of hundreds to thousands. APDs operating in the Geiger mode can trigger a self-sustaining collisional ionisation process even when a single photon is absorbed (generating a carrier pair), and the current will continue to increase until it is suppressed by an external resistor. Similarly, a pulse is generated when an additional one photon arrives. This shows that the Geiger-mode APD can only detect the presence or absence of photons and cannot characterise the intensity of the photons. The Figure 4 illustrates a schematic of how the APD works.

In addition, the optimisation of the key parameters of the APDs is essential to improve ranging accuracy. The critical parameters for APDs in satellite laser ranging (SLR) and lunar laser ranging (LLR) include the following:

Spectrum response: The material properties of the APD device in a single-photon detector determine the spectral response range of the detector. Different materials are used for different detection bands, e.g., SiC APDs for UV detection, InGaAs APDs for NIR detection, and Si APDs for visible light detection.

Multiplication Factor: This refers to the amplification of the final output photocurrent relative to the initial photocurrent after the initial photogenerated carriers undergo mechanisms such as collisional ionisation during the avalanche process. The multiplication factor, M, has been observed to increase rapidly as the applied reverse bias voltage, *V*, approaches the breakdown voltage, $V_{B}$. At the point of *V* = $V_{B}$, the value of *M* is +*∞*, and the gain saturation effect ensues. The relationship between these two variables can be approximated by the following equation:(2)$$M=\frac{1}{1-{\left(\frac{V}{V_{B}}\right)}^{n}}$$
where n is a constant, determined by the semiconductor material, doping distribution, and wavelength of incident light.

Dark current: The presence of false counts at the detector output in the absence of incident photons is attributable to two primary factors: thermal noise and quantum tunnelling effects within the APD device. This parameter is related to the size of the photoreceptor surface of the APD device, its operating temperature, and the bias voltage applied to its terminals.

Timing jitter: In the process of incident photons being absorbed by the photosensitive surface of the APD to the output of the electrical pulse signal from the detector, the output time delay shows an uncertain distribution. The utilisation of the TCSPC counting apparatus, the return photon triggered by the electrical pulse signal measurement, and the cumulative period of time will be in the time domain to form a counting time distribution, showing a Gaussian-like distribution of the counting envelope; the FWHM of this envelope is the detector’s time jitter. Within the field of laser ranging applications, the temporal variation in the light signal’s propagation is of paramount importance in determining both the accuracy and precision of measurement, as well as the distance resolution capability of the ranging system.

Detection efficiency: This is the probability that an incident photon is successfully detected, expressed as the ratio of the photon count value of the detector to the number of incident photons.

These parameters are crucial for detecting weak return signals (photon counts down to picosecond pulses of a single photon) while minimising noise and timing errors. It is worth noting that avalanche gain itself has a significant temperature dependence: An increase in temperature leads to a decrease in the collisional ionisation probability and an exponential decay of the gain coefficient (*M*) with temperature, which directly affects the stability of the gain bandwidth product. To optimise these parameters, advances have focused on device structure design (e.g., separate absorber, graded, charge and multiplier layers), materials engineering (e.g., the use of GeSi or III–V compound semiconductors to improve sensitivity), process optimisation (e.g., precise doping distribution and defect control to reduce temperature-induced tunneling noise and dark current surges), and circuit optimisation (e.g., time travel compensation combined with temperature feedback circuits to counteract temperature drift by dynamically adjusting bias or screening thresholds).

### 3.1. Si-APD Research Progress and Application

The 532 nm laser is the main light source used in laser ranging systems, featuring high brightness, high atmospheric transmittance, safe visibility to the human eye, and high compatibility with current optical instruments. Among many semiconductor material systems, Si has a band gap of 1.1 eV, which is one of the ideal choices for detecting weak optical signals in the 400–1100 nm band [18]. Therefore, Si-APD is extremely widely used in conventional SLR systems.

In 1986, C.O. Alley et al. of the University of Maryland, USA used the RCA C30902E Si-APD with a quantum efficiency of 20% for satellite laser ranging, and the ranging error reached 15–20 mm [19]. In the same year, CTU successfully developed Si-APD with a quantum efficiency of 20% and a time shift of about 20 ps. However, the diameter of the photosensitive surface is only 0.1 mm, the optical path alignment requirements are high, and the noise is relatively high [20]. The device was successfully applied to the Graz and Helwan stations, obtaining centimetre-level SLR observations [21]. Subsequently, the CTU team, represented by Prof Ivan, further improved the performance of the Si-APD by adopting an active gated quenching circuit and tested it at the Graz station in Australia, the Shanghai Observatory of the Chinese Academy of Sciences (CAS), the RGO station in the UK, and the MTLRS-1 system in Germany, with a systematic error drift of only about 5 ps. The ranging system temporal stability is demonstrated on Figure 5 [22].

In 1994, G. Kirchner et al. found that Si-APD suffered from severe time jitter effects at multiphoton incidence. When the number of incident photons exceeded 1000, the time jitter exceeded 200 ps, resulting in a measurement error of 30 mm. Subsequently, G. Kirchner et al., based on the time jitter effect and the number of incident photons, proposed that the discriminator adopts a bi-level mode of operation, the use of time wander compensation circuits of the APD, the successful development of C-SPAD K14 with a time wander compensation function and package thermostat control, so that the time jitter of the device is reduced from 200–300 ps to less than 20 ps, and the temperature characteristic is −0.6 mm/°C [23]. Figure 6 [24] shows the structure of the C-SPAD circuit and Figure 7 [25] shows the results of the comparison before and after the compensation for the time delay.

In 2007, CTU conducted a study of the 200 μm K14 SPAD on the target surface. The results show that the time jitter of the K14 SPAD is only 40 picoseconds and that the response of the photosensitive surface has good uniformity, with detection delay fluctuations of only a few picoseconds even at the edges in the 200 μm diameter range [26]. The relative sensitivity, time jitter, and relative detection delay characteristics of the K14 diode are shown in Figure 8.

To further improve the performance of the K14 SPAD, the CTU team, in 2013, proposed a passive circuit compensation technique based on the relationship between the time jitter and photon number, which effectively reduces the RMS of the Si-APD time resolution to 20 ps and achieves a photon detection efficiency of more than 40% in the wavelength range of 500 nm–800 nm. The temperature dependence of the output delay of this APD is only 280 fs/K under the temperature change conditions from 20 °C to 50 °C. In 2017, the CTU team upgraded the compensation circuit for the second time with the active gated quenching technique to obtain higher time stability. The temperature drift coefficient was reduced to 70 fs/K over the same temperature variation range [28].

In 2020, the CTU team developed a Si-APD with a detection accuracy of 8.6 ps RMS and time stability of 0.3 ps@300 s. Additionally, the device demonstrated a temperature drift coefficient as low as 6 fs/K between 15 and 55 °C, as well as a stable detection delay of ±1.5 °C over a wide temperature range of −55 °C to +55 °C [29]. In 2022, based on the K-14 SPAD with a diameter of 100 μm, by introducing an ultrafast comparator and optimizing the related signal logic, the sensitivity of the detection delay to temperature changes will be greatly reduced, and the temperature drift coefficient will be reduced from 250 fs/K to 100 fs/K, providing technical support for laser time comparison and ranging [30]. In addition to the CTU team, the Italian company MPD has also been responsible for the development of a high-precision Si-APD in recent years. The device is capable of achieving a detection accuracy of 12–15 ps with low dark counts [31]. Nevertheless, MPD has yet to develop a high-stability readout circuit to complement this APD detector, which is currently unable to satisfy the demand for high-stability detection.

The mass production of Si-APD devices has historically been dominated by developed countries, with notable examples including First Sensor of the United States and Hamamatsu of Japan. The field of Si-APD research in China is still in its nascent stages. In 2018, the Beijing University of Posts and Telecommunications (BUPT) developed a high-precision Si-APD device with a target surface diameter of 200 μm. In 2019, East China Normal University (ECNU) proposed Si-APD for high-precision satellite laser ranging applications, with a time stability of 0.15 ps@100 s, a single-measurement accuracy of 22 ps, and an output delay drift of only ±1 ps over the temperature range of 16 °C–36 °C [32]. Furthermore, the Institute of Semiconductor Research of the Chinese Academy of Sciences, the Southwest Institute of Technical Physics, the 209 Institute of Weapon Industry Group, and the 44th Institute of China Electronic Science and Technology Group have conducted research on Si-APD. In recent years, the 44th Research Institute of China Electronics Technology Group Corporation (CETC) has been engaged in the promotion of Si-APD devices for military-to-civilian applications in small quantities. Nevertheless, due to confidentiality and other considerations, there is a paucity of publicly available reports.

### 3.2. InGaAs/InP APD Research Progress and Applications

In recent years, there has been a gradual increase in interest in 1064 nm band near-infrared laser ranging technology within the industry. In comparison to the 532 nm visible light band, the number of photons emitted by a 1064 nm laser can be increased to twice the original level under the same single-pulse energy conditions. This effectively reduces the detection difficulty of the system and improves the detection distance. Furthermore, in accordance with the theory of atmospheric scattering and absorption, the short-wave infrared band exhibits a higher atmospheric transmission rate than the visible light band. This is particularly evident in low-elevation observation conditions, where the atmospheric transmission advantage is pronounced. Furthermore, in daytime ranging, the 1064 nm laser exhibits a sky background noise intensity that is one order of magnitude lower than that of the 532 nm band. It is evident that near-infrared laser ranging technology exhibits considerable advantages in terms of signal-to-noise ratio and transmission distance.

In APD applications in the 1064 nm band, the InGaAs(P)/InP material system is favored due to its high absorption efficiency in the near-infrared region. In 1978, Pearsall and Papuchon [33] designed InGaAs APDs using a tabletop type with a homogeneous junction structure, but the tunnelling current was extremely high due to the small forbidden bandwidth. In order to solve the above problems, Taguchi et al. [34]. proposed a heterojunction APD (separated absorption and multiplication APD, SAM APD) in which the absorption and multiplication layers are separated. The incorporation of the InP multiplication layer facilitates the generation of a sufficiently high electric field, which, in turn, enhances collisional ionisation and achieves high avalanche gain. This improvement also serves to enhance the quantum efficiency of the APD. The SAGCM (separate absorption grading charge multiplication) structure is further optimised on the basis of SAM through the design of a gradual electric field distribution, which serves to control the avalanche process, reduce the surface leakage current, and improve the uniformity of the multiplication process [35]. This results in an improvement in the APD gain and noise reduction. The electric field distribution is shown in Figure 9.

The diffusion of the p+ region is a critical step in the fabrication of SAGCM-structured InGaAs/InP APDs. p+ region diffusion profiles are ideally rectangular. However, due to the anisotropy of the diffusion process, the resulting diffusion region tends to be curved, which leads to premature breakdown at the junction edges. To solve this problem, Kao and Wolley [36] proposed the concept of a floating guard ring (FGR). The FGR is implemented at a specific distance from the main junction region with the objective of creating a new p+ region through supplementary diffusion. The region is situated in close proximity to the main junction, which has the effect of reducing the surface electric field, decreasing the device leakage current, and ensuring the stability of the SAGCM-structured InGaAs/InP APD devices.

InGaAs/InP APDs were first used at AT&T Bell Labs in 1985 [37]. In 1994, Lacaita [38] employed gating and cooling techniques to achieve a noise-equivalent power, along with a time resolution of less than 1 ns and an optimal full width at half maximum (FWHM) of 200 ps. In 2002, K. A. McIntosh et al. at MIT Lincoln Laboratory undertook the development of a 1064 nm InGaAsP/InP APD, resulting in the creation of an APD with a detection efficiency of 33% and a dark count rate of 1.7 MHz at 290 K [39]. In 2006, Joseph P. Donnelly et al. proposed a theoretical model for the variation of dark count rate and detection efficiency in 1064 nm InGaAsP/InP avalanche photodiodes (APDs) with overbias voltage and temperature, based on experimental data. For an APD with a 10 μm diameter active region, a DCR of less than 10 kHz can be achieved at room temperature with a 5 V overbias voltage and a 50% PDE [40]. In 2007, Mark A. Itzler et al. [41] used InGaAsP as the 1064 nm APD absorber layer, and by optimising the internal electric field distribution, achieved 10% detection efficiency at −40 °C with a dark count rate below 1 MHz, opening up new possibilities for single-photon detection applications. Xudong Jiang et al. [42] used numerical simulation to analyse the relationship between the InGaAsp/InP dark count rate and the detection efficiency, and analysed the methods to optimise the performance of the APD in different modes; when the dead time was increased from 200 ns to 460 ns in the free-running mode, the DCR was decreased by 50% (250 K) in the gated mode, and the detection efficiency was increased to 35%. In the same year, Verghese of MIT Lincoln Laboratory put forth the proposal of 8*8 and 32*32 arrays with 100 μm spacing, 128 × 32 and 256 × 64 arrays with 50 μm spacing, and the utilisation of loaded quenching circuits for InP APDs with frame readout and pixel auto-reset [43]. The findings demonstrate that the detection efficiency of InGaAsP/InP reaches 50% at room temperature (1.06 μm), with a dark count rate of 20 kHz and a dead time of 6 μs. In 2008, David A. Ramirez et al. [44] put forth a methodology to enhance the APD detection efficiency by optimising the width of the doubling region and the overbias voltage. This was achieved by analysing the relationship between the performance of the 1064 nm APD and the width of the doubling layer, as well as comparing the effects of field-assisted attempts to penetrate and temperature-assisted dark carriers. In 2011, Mark A. Itzler et al. [45] achieved a 20% detection efficiency while acquiring a time jitter of less than 50 ps based on a self-quenching design with monolithic integrated thin film resistor feedback. Subsequently, Mark A. Itzler et al. proceeded to optimise the InGaAsP/InP APD structure and simulation parameters with the success of achieving a dark count rate that was two orders of magnitude lower than that of the 1550 nm APD, while maintaining a higher detection efficiency. Figure 10 [46] illustrates the dark count rate and detection efficiency of the InGaAsP/InP APD with an 80 μm diameter at 1.06 μm, as a function of temperature.

In addition to the research conducted at the MIT laboratory, the Italian Cova group and the CTU Ivan team are engaged in the investigation of 1064 nm APD. A number of companies worldwide are also gradually increasing their research efforts in the field of InGaAs(P)/InP near-infrared APDs. Examples of such companies include Princeton Lightwave, whose products include the PGA series, Thorlabs, which offers the SPCM series, and ID Quantique, which provides the ID230 series. China’s research institutions on near-infrared APDs include the 44th Institute of CETC (Chongqing Institute of Optoelectronic Technology), the 11th Institute of CETC (North China Institute of Optoelectronic Technology), the Kunming Institute of Physics (China Ordnance 211 Institute), the 209th Institute of China Ordnance Industry Group (Southwest Institute of Physics), the China Air-to-Air Missile Research Institute (The AVIC 612th Institute), the Shanghai Institute of Technical Physics of the Chinese Academy of Sciences, East China Normal University, the University of Science and Technology of China, Nanjing University, the Changchun Satellite Observatory of the Chinese Academy of Sciences, and other institutions. In recent years, Li Bin’s team at the 44th CETC Institute has studied the effect of the preparation process of InGaAs/InP near-infrared APDs on the device. The 13th CETC Institute and the Changchun Institute of Optics have made significant progress in near-infrared APD research. However, compared with commercial APD detection devices in developed countries, China’s research is relatively backward.

Currently, 1064 nm laser ranging has been achieved at some observatories. The application of this technology is not only limited to satellite ranging, but has also been successfully extended to the accurate measurement of the retroreflector array. In 2005, the Wrightwood (OCTL) Observatory in the United States began detecting laser echoes at a wavelength of 1064 nm using an InGaAs APD (model PDA 400; Thorlabs Inc., Newton, NJ, USA). In 2011, A. McCarthy et al. presented the first single-photon laser ranging using quasi-continuous detection with high-speed gated InGaAs/InP APDs [47]. The technology markedly enhanced the detector’s performance through a high-speed gating method, thereby facilitating single-photon detection and ensuring high precision in laser ranging applications. In 2014, Huang et al. investigated the potential of InGaAs/InP avalanche photodiodes for human eye-safe laser ranging and successfully applied them to this purpose. In 2016, the Yunnan Station of the Chinese Academy of Sciences employed a 1064 nm band laser in conjunction with superconducting single-photon detectors. Laser ranging was achieved for cooperative targets. Following the implementation of system optimisations, near-infrared laser ranging for space debris was achieved in 2017 [48]. In the same year, Agata M. Pawlikowska et al. conducted time-dependent single-photon-counting experiments utilising an InGaAs/InP APD, acquiring depth and intensity profiles of targets across a range of 10 km and achieving high-resolution three-dimensional imaging [49]. In 2021, the research group led by Jianwei Pan at the University of Science and Technology of China (USTC) successfully addressed the challenges of atmospheric attenuation and background noise by optimising the Cassegrain telescope optical system and the low-noise InGaAs/InP APD. This enabled the completion of an ultra-telephoto imaging experiment with a range of 200 km and the realisation of a three-dimensional image reconstruction at very low signal strength [50]. In 2022, Guangyue Shen et al. employed a GHz-gated Geiger-mode InGaAs/InP APD in conjunction with a galvanostat scanning technique to achieve the generation of large-range single-photon images. The system was capable of generating an effective point cloud comprising approximately 1.3 × 10^5^ points in a single second, enabling the successful imaging of targets situated at distances ranging from 200 m to 1200 m [51]. In the same year, R. Bimbova et al. [52] developed two compact photon-counting detectors based on the PGA-200-1064 InGaAs/InP APD detector chip, which was based on passive and active quenching principles. Among them, the APD with passive gating circuit has a detection probability of up to 80% within a 100–500 ps gate width, with a time resolution of only 28 ps RMS; the APD with active gating circuit has a detection probability of 70% within a 50 ns–10 μs gate width, with a time resolution of 67% and a time stability of better than 0.2 ps at 5000 s of averaging, which is suitable for applications that need sub-picosecond long-term detection delay stability, such as space debris optical tracking.

In 2023, the Changchun Observatory successfully researched and developed the 1064 nm APD by adopting passive gated quenching technology through independent research and development. This has enabled the Observatory to meet the needs of daytime near-infrared laser ranging technology. The device employs the SAGCM APD chip structure, exhibiting an 80% quantum efficiency, a 1.17 A/W response, and a 40 ps time jitter. Figure 11 illustrates the 1064 nm APD developed by Changchun Observatory.

In 2024, Changchun Observatory constructed a 1.2 m near-infrared space debris laser ranging (DLR) system based on the above devices and carried out near-infrared laser ranging experiments in Cairo, Egypt, obtaining eight laps of test data for cooperating targets, among which, four laps were for cooperating targets with the maximum number of points obtained in a single lap of 51,479 points and four laps were for non-cooperating targets with the equivalent minimum size of 0.6 m@1000 km. In the same year, the first daytime near-infrared space debris ranging was achieved, with 1738 effective points. Figure 12 illustrates the observation interface of the 1.2 m near-infrared DLR system.

In comparison to Si-APDs, InGaAs(P)/InP APDs exhibit notable advantages in terms of detection efficiency and avalanche gain within the near-infrared band. Nevertheless, further improvements are required in the dark count rate, time resolution, temperature characteristics, and fabrication cost of the devices due to the inherent defects in the InGaAs(P)/InP material and the deficiencies in the semiconductor process. It is worth noting that GeSn and Ge-on-Si materials show potential in the near- to mid-infrared wavelength bands by virtue of their compatibility with the CMOS process [53,54]. Although there is currently less research in the field of satellite laser ranging, their low cost and integration advantages are expected to drive future technological development.

### 3.3. Avalanche Photodiodes in Mini SLR Systems

In recent years, Daniel Hampf [55] has proposed miniSLR technology, which is a compact and transportable SLR system that facilitates the application of SLR technology to new projects by reducing the cost of ownership and simplifying the site establishment process. The system is illustrated in Figure 13.

In the design of the mini SLR system, the 532 nm green light after frequency doubling is currently utilised as the operating wavelength, primarily due to the dependence of single-photon detection technology on the visible spectrum. The silicon-based Geiger mode avalanche photodiode (Si APD) demonstrates picosecond timing accuracy and high detection efficiency within the 532 nm band; nevertheless, its spectral response range imposes constraints on the system’s laser wavelength selection. However, recent advancements in InGaAs APD technology, such as picosecond timing accuracy and low dark count rate, have led to significant improvements in performance. Concurrently, the chip size of the new InGaAs APD has been reduced to the millimetre level, offering substantial advantages in terms of compactness and integration. This development enables miniSLR systems to directly adopt the fundamental frequency wavelength (1064 nm) of Nd: YAG lasers, thereby eliminating the frequency doubling module. This, in turn, significantly simplifies the optical structure and reduces photon loss, directly promoting the compactness of laser emission modules. Furthermore, hybrid integration schemes based on silicon photonics platforms, such as InGaAs APD coupled to silicon waveguides, are expected to enable the on-chip integration of detectors and signal processing units [56], laying the foundation for a new generation of “miniSLRs”. In addition, the passive quenching active reset (PQAR) circuit recently proposed by Hsi-Hao Huang et al. [57] provides a significant reference point for the highly integrated, high-precision, low-noise, and high-repeat-frequency single-photon detectors that are essential for the mini SLR. The circuit is notable for its capacity to reduce the photon signal jitter to 63 ps (FWHM) through the utilisation of a self-restoring logic control circuit. This development signifies a substantial enhancement in time measurement accuracy and facilitates the sub-millimetre resolution of time-of-flight (ToF) in laser ranging. Concurrently, the PQAR circuit exhibits a dead time of merely 5.46 ns during continuous photon events, which, when coupled with a theoretical maximum count rate of up to 200 MHz, fulfils the requirement for expeditious response and data acquisition rates for high-frequency laser ranging. In comparison with conventional passive quenching passive reset (PQPR, jitter 1.7 ns) and passive quenching active clock-driven reset (PQACR, jitter 710 ps) circuits, PQAR represents a significant advancement in terms of time resolution and anti-interference capability. At the process level, the 0.18 μm HV CMOS-based APD design supports highly integrated arraying. However, for the mini SLR system, the high voltage requirement (bias voltage of 52 V) may pose challenges in terms of power consumption and integration, and remains to be further optimized.

## 4. Trends in Laser Ranging Avalanche Photodiode Research

Laser ranging technology is one of the technologies with the highest precision for single-point measurement of satellites, and plays an important role in the precision orbiting of targets, monitoring of space debris, determination of the Earth’s rotation parameters, and establishment of the global Earth reference frame and maintenance. As the range of applications continues to expand, the demand for enhanced accuracy and stability in laser ranging technology is growing. The International Laser Ranging Service (ILRS) has explicitly defined the requisite accuracy for ranging measurements of low Earth orbit (LEO) satellites as better than 1 cm, while for geosynchronous orbit (GEO) satellites, the accuracy should be better than 5 cm. Furthermore, the Global Geodetic Observing System (GGOS) has set forth a requirement of 0.1 mm/year for the stability of SLR data.

In order to meet the demands of laser ranging in terms of high accuracy, long range, and intelligence, single-photon detectors must have a higher detection efficiency in order to ensure that every incident photon can be accurately detected. Concurrently, the device must also diminish the time jitter in order to enhance the precision of time measurement and diminish the impact of background light and stray light on the measurement outcomes by reducing the dark count rate. In order to meet the diverse requirements of potential applications, laser ranging technology must also make significant advances in miniaturisation and lightweighting. This will necessitate the development of a more compact design of single-photon detectors, facilitating their integration into a range of devices, including unmanned aerial vehicles (UAVs) and mini SLR systems. Furthermore, single-photon detectors must exhibit enhanced stability and reliability to ensure optimal performance in extreme environments. They must be capable of functioning normally under harsh conditions, including high temperatures, low temperatures, high humidity, and strong vibrations. In order to achieve these goals, the further optimisation and improvement of laser ranging single-photon detectors can be pursued in the following ways.

The sensitivity and response speed of single-photon detectors are directly affected by the materials used to construct them, particularly in terms of their semiconductor properties. The adoption of advanced epitaxial technologies, such as molecular beam epitaxy (MBE) and metal-organic chemical vapour deposition (MOCVD), enables the development of high-purity epitaxial materials, the reduction of defects and impurities, and, thus, the reduction in the device’s time jitter and dark count rate. This, in turn, leads to an effective improvement in the detection accuracy of laser ranging. Concurrently, the advancement of energy band engineering and epitaxial growth technology for semiconductor materials has led to an expansion in the range of materials available for use in laser ranging single-photon detector chips. By modifying the proportion of multiple compound semiconductor materials, the band gap width can be modified to optimise the material’s optoelectronic properties, thereby enhancing the detection efficiency, response speed, and temperature characteristics of specific wavelength photons. For instance, InGaAs and InGaAsSb possess band gaps that are well-suited to infrared wavelengths, rendering them highly effective in near-infrared single-photon detectors for lunar laser ranging, space debris laser ranging, and other applications.

The application of avalanche photodiode (APD) arrays significantly improves the detection performance, but still faces the two challenges of pixel crosstalk and performance consistency. To address the crosstalk problem, dynamic gating control technology can be used to suppress the interference in the non-operating region, combined with the tabletop isolation structure and three-dimensional integration process to reduce the optical/electrical coupling effect, and deep learning algorithms can be used to enhance the signal recognition ability under a low signal-to-noise ratio. In performance consistency optimisation, the broad spectrum material system and precision epitaxial process improve the response uniformity, and the dynamic bias compensation and screening matching technology are expected to further enhance the array stability. Future technology will focus on the direction of ultra-large-scale array, single-photon level sensitivity, and intelligent self-calibration, but it needs to break through the bottlenecks of material epitaxy and process precision to support multi-scene applications.

In terms of chip structure, optimising the chip structure design will help to improve detection efficiency and reduce noise levels. For instance, the incorporation of a photonic crystal structure can facilitate the regulation of the detector’s light absorption and emission characteristics. The implementation of a multi-layer heterogeneous structure can enable the selective detection of specific wavelengths of optical signals. The utilisation of quantum dots, nanowires, and other innovative chip structures can enhance the photoelectric conversion efficiency and signal amplification of the detector, thereby improving the detector’s sensitivity and response speed. The incorporation of a grating structure or reflective layer serves to enhance the APD’s capacity to capture incident photons, thereby improving the detection efficiency. Furthermore, the utilisation of miniaturised optical components, such as microlens arrays, is a potential avenue for exploration. This not only reduces the mass and volume of the detector, but also maintains or improves the detection sensitivity, which provides crucial support for the deployment of laser ranging technology in satellite and airborne platforms.

In regard to peripheral driving circuits, the key to enhancing the performance of laser ranging single-photon detectors lies in attaining rapid response, self-adaptation, minimal energy consumption, and high integration. The combination of passive gated bias circuits, active gated bias circuits, and digital signal processing techniques can effectively reduce time jitter and noise, thereby improving the accuracy and speed of detection of the device. Furthermore, the incorporation of temperature or optical sensors, in conjunction with micro- and nanocontrollers (e.g., Arduino, Raspberry Pi, etc.) for data acquisition, and the establishment of an environmental feedback control system, enables the bias circuit to autonomously adjust the operating parameters, thereby facilitating adaptation to a broader range of environmental conditions. Furthermore, the utilisation of low-power operational amplifiers and high-efficiency power management circuits, in conjunction with the integration of bias circuits, signal amplification, filtering, and other functions into a single chip, will not only result in a reduction in the overall energy consumption of the system, but it will also facilitate an enhancement in the compactness and integration of the device, thereby enabling the fulfilment of the performance requirements of the portable laser distance measurement system.

In the context of signal processing, the development of high-efficiency, real-time, and high-precision single-photon detector signal processing methods represents a crucial technology for the realisation of ultra-high frequency laser ranging. Furthermore, the utilisation of hardware acceleration technologies, such as field-programmable gate arrays (FPGAs) or graphics processing units (GPUs), can enhance the speed and efficiency of signal processing. Additionally, the investigation of deep learning algorithms can be conducted through the employment of artificial neural networks. A multilayer network structure is established based on the different time-domain characteristics of the echo effective signal and the noise signal. This structure is trained using a large amount of data, which is used to discriminate and extract features from the data acquired by the single-photon detector. This process improves the accuracy and speed of data processing, allowing for the rapid identification and classification of the massive laser ranging data.

## 5. Conclusions

Laser ranging technology represents a high-precision geodetic tool that is inextricably linked to matters of national security, economic development, and scientific research. In accordance with the operational principles of laser ranging and the technical specifications, this paper categorises and synthesises the operational mode, technical parameters, and utilisation status of diverse photodetectors. In comparison to other photodetectors, the Geiger-mode APD exhibits distinct advantages in the domain of high-frequency and high-precision laser ranging. By examining the advancements and applications of prevalent single-photon detectors in laser ranging, such as Si-APD and InGaAs/InP APD, this paper elucidates the trajectory of single-photon detector development in the domain of laser ranging. The advancement of integration technology, intelligent algorithms, and environmental adaptability technology will facilitate the development of high-performance APDs, which will, in turn, provide stronger technical support for the advancement of ultra-high-frequency and ultra-high-precision laser ranging technology.

## Figures and Tables

**Figure 1 sensors-25-02802-f001:**
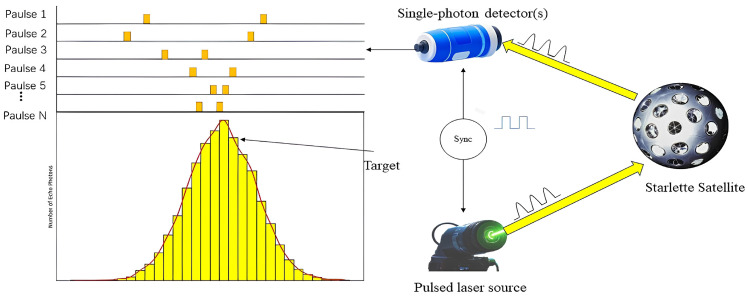
Distribution of time-dependent photon counts.

**Figure 2 sensors-25-02802-f002:**
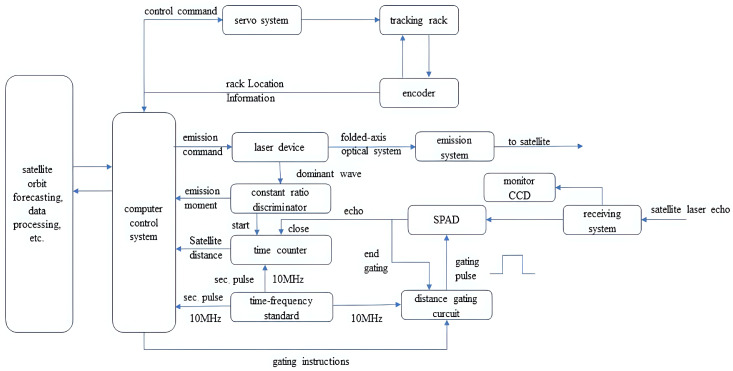
Block diagram of SLR system at Changchun station.

**Figure 3 sensors-25-02802-f003:**
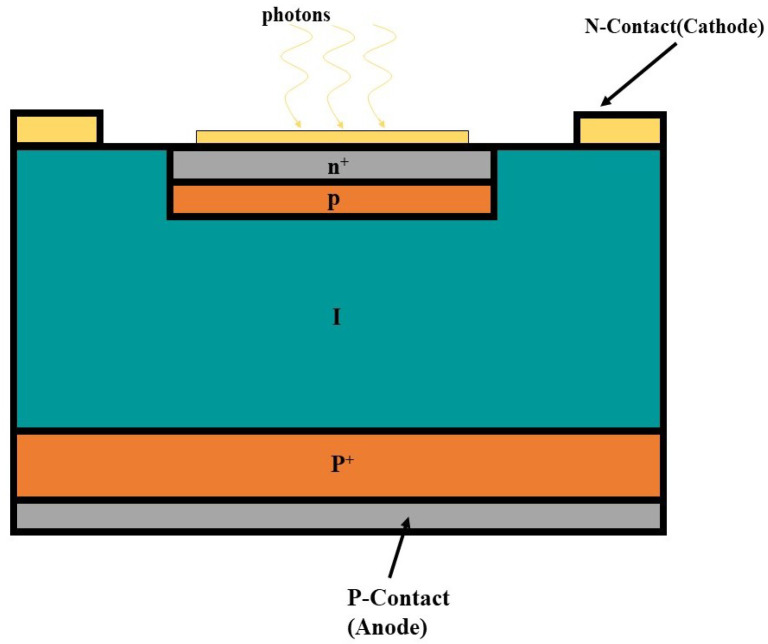
Typical avalanche diode structure diagram.

**Figure 4 sensors-25-02802-f004:**
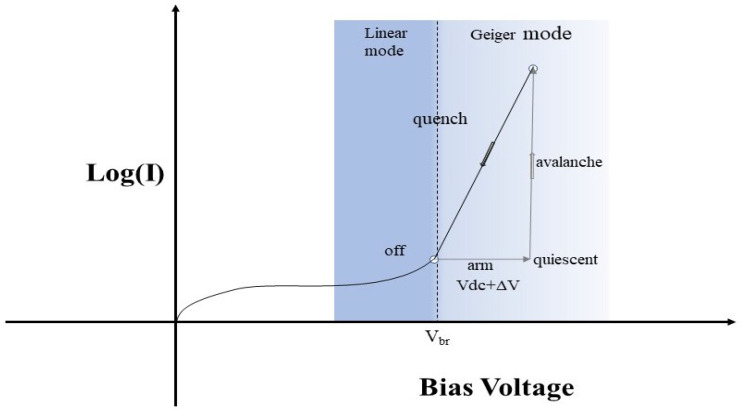
Typical operating IV curve for APDs.

**Figure 5 sensors-25-02802-f005:**
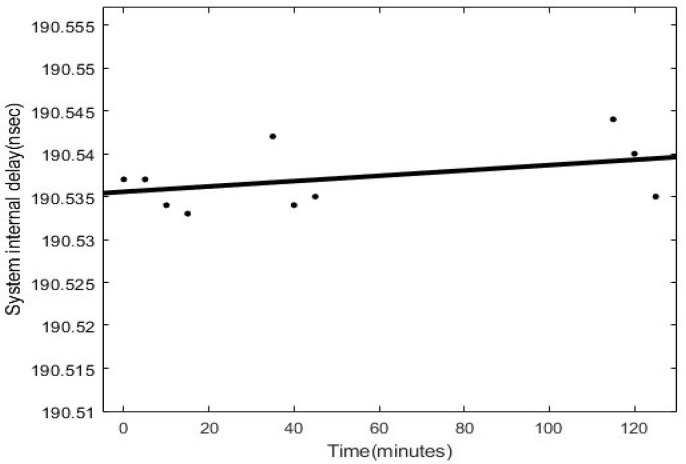
SLR system temporal stability.

**Figure 6 sensors-25-02802-f006:**
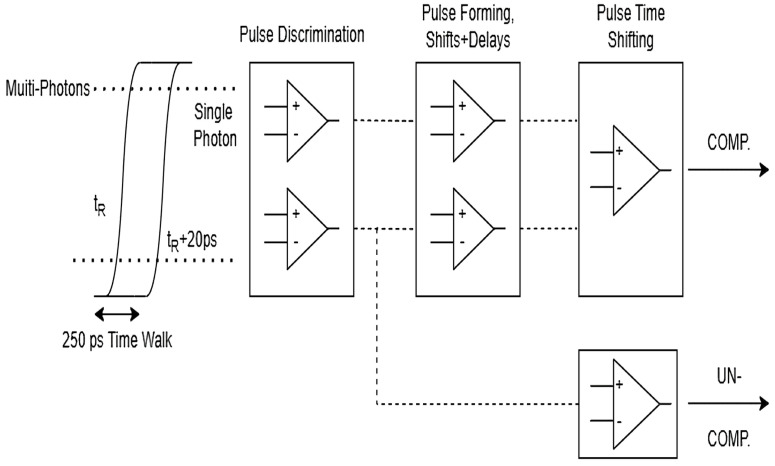
Time walk compensation: electronic circuit schematics.

**Figure 7 sensors-25-02802-f007:**
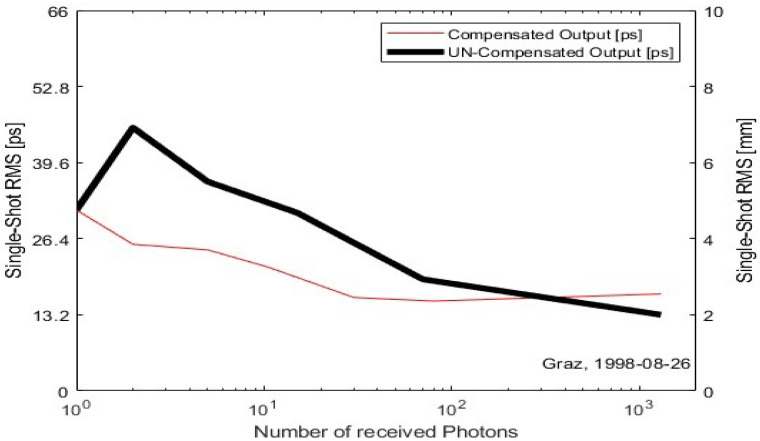
Comparison of time walk compensated vs. uncompensated C-SPAD K14.

**Figure 8 sensors-25-02802-f008:**
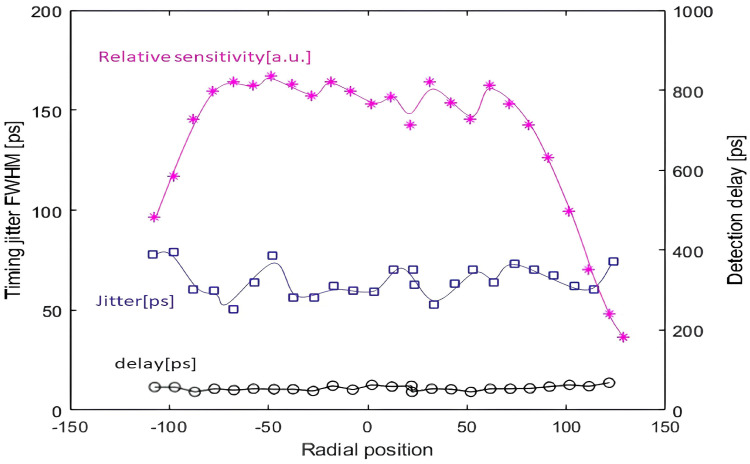
Characteristics of a typical 200 µm K14 SPAD [27].

**Figure 9 sensors-25-02802-f009:**
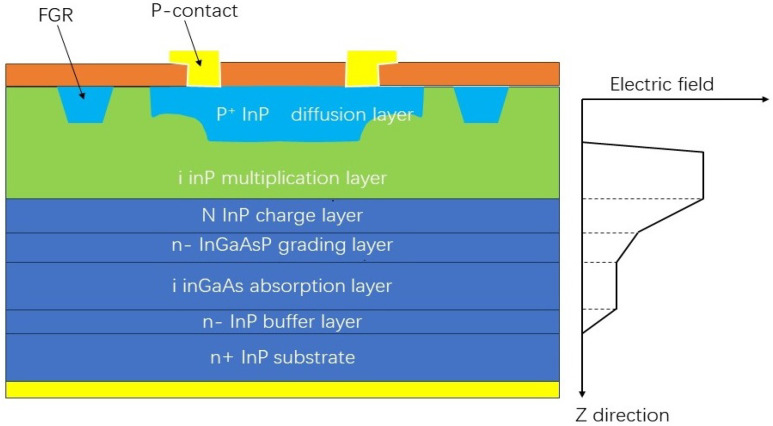
SAGCM structure and its internal electric field.

**Figure 10 sensors-25-02802-f010:**
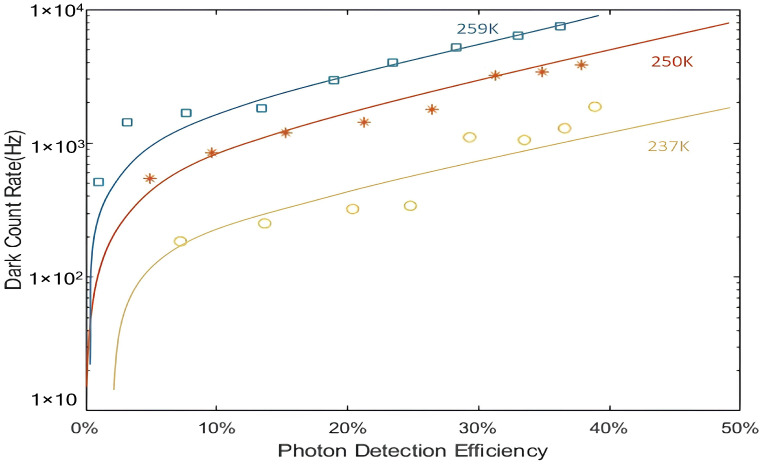
Relationship between dark counting and photon detection efficiency at different temperatures (1.06 μm wavelength, 80 μm diameter).

**Figure 11 sensors-25-02802-f011:**
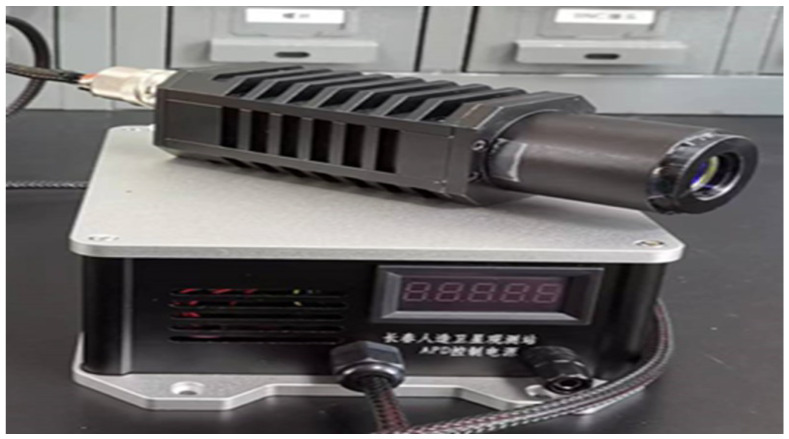
InGaAs/Inp APD developed at Changchun Observatory.

**Figure 12 sensors-25-02802-f012:**
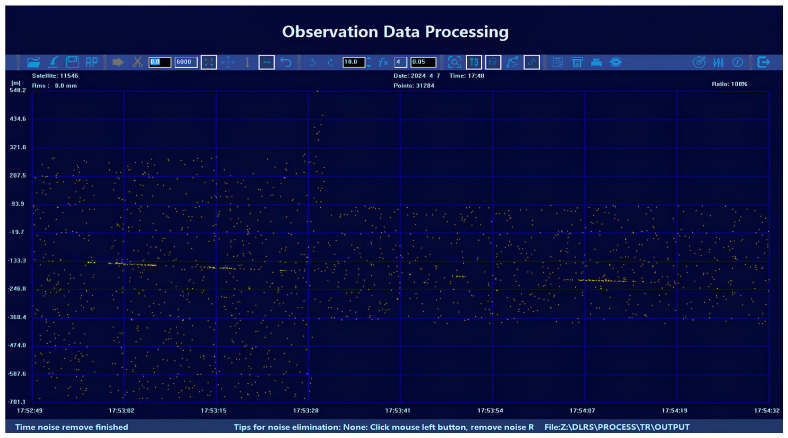
Near infrared DLR system observation interface.

**Figure 13 sensors-25-02802-f013:**
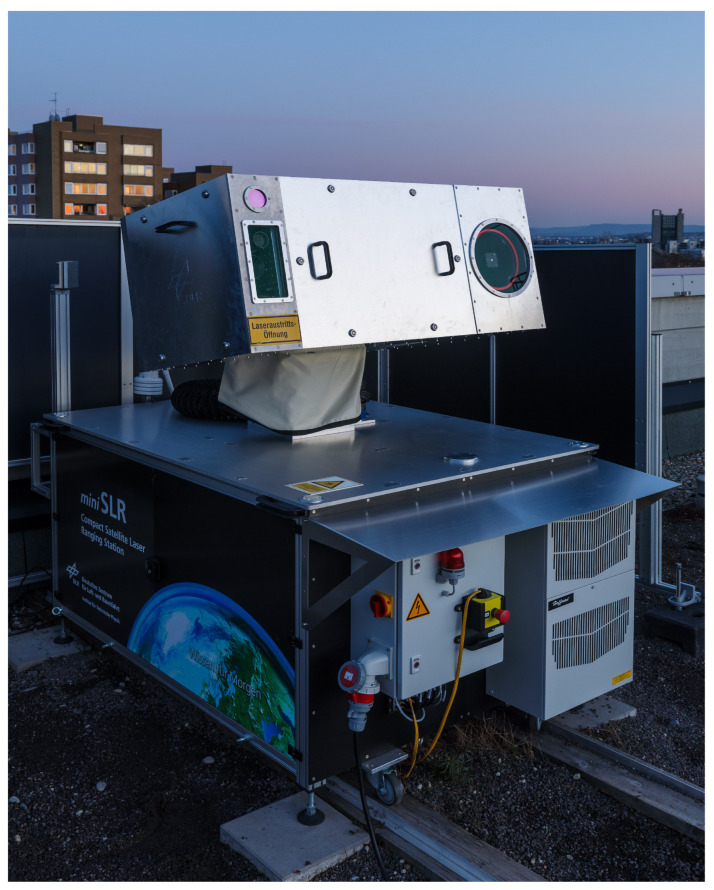
The mini SLR prototype on the roof of the DLR institute building. The enclosure in the bottom contains most of the electronics and IT. The top compartments house the receive and transmit telescopes, laser head, cameras, detectors, and beam control optics.

**Table 1 sensors-25-02802-t001:** Measurement of the number of echo photons from different satellites at Changchun Observatory.

Target Satellite	Laser Type and Wavelength (nm)	Detector Type and Applicable Wavelength (nm)	Mean Time of Light (s)	Perigee Altitude (km)	Zenith Angle (°)	Return Photon Number
ajisai	Nd-Yag@1064	CSPAD@532	0.017116485598	1479	54.82660396144291	118.9024
lageos-2	Nd-Yag@1064	CSPAD@532	0.041059333398	5617	24.21489660834258	3.5909
etalon-2	Nd-Yag@1064	CSPAD@532	0.132175862032	19097	15.58953384198299	0.033438
beidou3m-1	Nd-Yag@1064	CSPAD@532	0.147367634768	21519	13.22384987727686	0.021639

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
