# Peer review of "Progress in Avalanche Photodiodes for Laser Ranging"

_sensors, 2025, doi:10.3390/s25092802_

Round 1
Reviewer 1 Report
Comments and Suggestions for Authors
The present work is a review of avalanche photodiode (APD). The selected topic is interesting, but the target looks unclear to me (comment 3). In the review paper, topic and selected contents should be in the same direction. Below are some comments.
- The comment “APDs operating in linear mode are unable to detect single photons, although they have a high sensitivity and have been able to achieve sensitivities of 500-1000 photons” is unclear. In my field, detection of several photons is possible when the readout noise is reduced as much as possible.
- Typical avalanche gain should be commented. Further, huge temperature dependence of avalanche gain should be commented. Such properties are important both in fundamental science and device applications.
- The position/role of this article is unclear. In terms of basic science, basic mechanisms such as impact ionization (avalanche) are not touched at all. Regarding to device applications, examples look oriented to results of China. To my understanding/experience, leading technologies are come from US or Japan (Hamamatsu), as you comment in the paper. But device examples are mainly from China. If the title is R&D of APD in China or something, I agree the contents. But in the current style, title and contents are distinct. What is the purpose, general review of APD or Chinese R&D, on basic science or device properties?
- Relating to above comment, I know there are so many different names of Geiger-mode APD. Although you emphasize “single photon” as abbreviated to SAPD, detection of single photons is not explained well. If the style/logic is maintained in the current form, a common name of Geiger-mode APD should be used.
Reviewer 2 Report
Comments and Suggestions for Authors
This paper reviews the role of avalanche photodiodes (APDs) in laser ranging technology, which is valuable for readers interested in this field. However, several technical aspects need to be addressed for improvement.
(1) Although the paper presents a wealth of experimental data and technological advancements, it lacks a well - formulated, unified theoretical framework that comprehensively integrates all aspects of APDs in laser ranging. In-depth theoretical analysis would be invaluable as it would enable readers to better grasp the fundamental principles and the intricate relationships between various parameters.
(2) The description of the laser ranging system predominantly focuses on individual components, overlooking the critical aspect of how these components are integrated to achieve optimal performance. The alignment and synchronization between the laser, detector, and telescope play a pivotal role in determining the system's efficiency and accuracy. Misalignment or poor synchronization can result in signal loss, increased noise, and reduced ranging precision. In-depth discussion on system integration, including best practices, challenges, and potential solutions, is essential to provide a comprehensive understanding of laser ranging systems.
(3) Despite Si APD and InGaAs APD, there're also other APDs, such as, Ge APD, GeSn APD, which're also very promising CMOS compatible technology. So, it is highly recommended that authors should give a more compresensive review about this part. They're very important in this field, which can not be negleted. Please check the following refs.
Nature communications 10.1 (2019): 1086.
Nanomaterials 13.3 (2023): 606.
Sensors (Basel, Switzerland) 25.1 (2025): 263.
Nature 627.8003 (2024): 295-300.
IEEE Electron Device Letters (2024).
(4) In the context of mini-SLR systems and future applications, APD arrays are becoming increasingly relevant. However, the paper only briefly touches upon the potential of InGaAs SPAD arrays. To fully explore this topic, it should discuss various aspects such as crosstalk between array elements, the uniformity of performance across the array, and how to address these issues for seamless system integration. These factors can significantly affect the overall performance of APD arrays in laser ranging applications and, thus, require careful consideration.
(5) The future of APDs in laser ranging is likely to involve interdisciplinary research, such as the integration of optoelectronics with nanotechnology or quantum information science. However, the paper does not explore these potential interdisciplinary directions. Given the increasing importance of interdisciplinary research in advancing scientific knowledge and technological capabilities, this omission limits the paper's forward-looking perspective. A more comprehensive exploration of these interdisciplinary aspects could open up new avenues for research and innovation in the field of APD-based laser ranging.
Reviewer 3 Report
Comments and Suggestions for Authors
This a review of the development and current status of single photon APD technology in view of its application to laser ranging. Laser ranging is an important geodetic tool for satellite and debris ranging, lunar ranging as well as earth-bound applications. It requires detectors for very low intensities down to single photons, which high timing accuracy and low background.
The review clearly has an educational aim. It starts off with an account of laser ranging methodology, thus introducing the major figures of merit for signal photon detection (quantum efficiency, dark count rate, time jitter and resolution).
Section 3 deals with the history of SAPD technology and related electronics from the 1960s until today. Progress in Si and InGaAs semiconductors is retraced in detail with ample references. Visible as well as near-infrared wavelengths are covered. Section 4 provides an account of directions for likely future progress.
There are a number of inaccuracies which I list in the following:
p. 4, 3rd paragraph: The first sentence makes no sense.
p.5, 6th line: 500 m should be 500 µm. This is a recurrent error.
p.6, Figure 4: y-axis is log(I), not current. The word "waveform" is misleading, should be "curve"
p.6, Section 3.1, 4th line: the term "forbidden bandwidth" is misleading, should be "band gap"
p. 7, 3rd line: according to ref. 25, the measurement error is 30mm.
p. 7, 9th line: reference 26 would be more appropriate here.
p. 8, 1rst and 4th line: m should be µm in sensor dimensions
p. 9, 9th line: m should be µm in sensor dimensions
p. 10, Figure 9: "filed" should be "field" on the right plot
p. 10, 3rd line from the bottom: m should be µm in sensor dimensions
p. 11, lines 9 and 10: m should be µm in sensor dimensions
p. 11, line 12: m should be µm in wavelength
p. 11, Figure 10: reference is missing
p. 12, 2nd paragraph: lunar surface angle reflector is usually called a retroreflector array.
p. 15, line 24: m should be µm in sensor dimensions
p. 16, 10th line from the bottom: micro- and nanocontrollers (not micro and micro)
Round 2
Reviewer 2 Report
Comments and Suggestions for Authors
The authors have successfully addressed all previous concerns, transforming the manuscript into a high-quality, analytically rich review that balances comprehensive literature synthesis with original technical insights. The enhancements in comparative analysis, technical depth, and structural precision demonstrate a strong commitment to improving the work. Given the thoroughness of the revisions and the absence of remaining deficiencies, I recommend accepting this manuscript in its current form without further modifications.